

# How the blast-wave model describes PID hadron spectra from 5 TeV p-Pb collisions

Thomas A. Trainor

University of Washington, Seattle, USA

ttrainor99@gmail.com

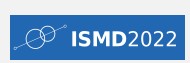

## Abstract

The blast-wave (BW) spectrum model has been applied extensively to nucleus-nucleus collision data with the intention to demonstrate formation of a quark-gluon plasma (QGP) in more-central A-A collisions. More recently the BW model has been applied to p-p, d-Au and p-Pb collisions. Such results are interpreted to indicate that "collectivity" (flows) and QGP appear in smaller systems. In this talk I review BW analysis of identified-hadron spectra from 5 TeV p-Pb collisions and examine the shape evolution of model spectra with collision centrality. I evaluate data-model fit quality using conventional statistical measures. I conclude that the BW model is not a valid data model.

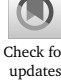

## 1 Introduction

The blast-wave (BW) spectrum model has been extended in recent years to identified-hadron (PID) spectrum data from small collision systems (e.g. *p-p*, *p*-A and *d*-A collisions at the RHIC and LHC). It is now conventional to interpret the existence of such BW fit results as confirming the presence of hydrodynamic flows in small systems and to infer quark-gluon plasma (QGP) formation as well [1]. As a result, the intended role of small systems as control experiments relative to QGP formation in more-central nucleus-nucleus or A-A collisions is vacated. In response, several questions emerge [2]: Can BW model fits actually demonstrate flows? Is there an alternative spectrum model with more likely physical interpretation that better describes spectrum data? What role do jets (nonflow) play in nuclear collisions? How should spectrum models be evaluated as to fit quality? And, is QGP actually formed in small collision systems?

## 2 Blast wave (BW) spectrum model

The BW model applied in Ref. [3] to 5 TeV *p*-Pb spectrum data is taken from Ref. [4] that introduced a hydrodynamics-based formula to describe pion spectra from fixed-target $\sqrt{s_{NN}} \approx 19$

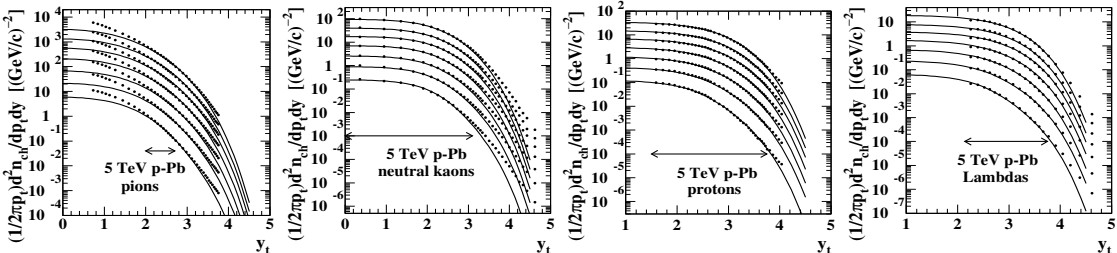

Figure 1: Data spectra (points) and BW model fits (curves) for four identified (PID) hadron species from 5 TeV $p$-Pb collisions [3]. Arrows indicate fit intervals.

GeV S-S collisions at the CERN SPS. The relevant formula is Eq. (7) of Ref. [4]

$$dn_{ch}/m_t dm_t \propto m_t \int_0^R r dr I_0[p_t \sinh(\rho)/T] K_1[m_t \cosh(\rho)/T], \tag{1}$$

with source boost $\rho = \tanh^{-1}(\beta_t)$, transverse speed $\beta_t(r)$ and mean transverse speed $\langle \beta_t \rangle$. $I_0$ and $K_1$ are modified Bessel functions. Equation (1) represents a thermal energy spectrum (Boltzmann exponential) in a boost (comoving) frame convoluted with a source boost (~speed) distribution on source radius to describe a particle spectrum in the lab frame.

Figure 1 shows BW fits (solid) based on Eq. (1) compared to PID spectrum data (points) from 5 TeV $p$-Pb collisions. The spectra are plotted as densities on $p_t$ (i.e. $p_t$ spectra as published) vs pion transverse rapidity $y_t = \ln[(p_t + m_{t\pi})/m_\pi]$ that provides improved visual access at low $p_t$. Model curves are generated using fitted BW parameters from Table 5 of Ref. [3]. The arrows indicate fit intervals for each hadron species. Substantial data-model deviations are notable.

## 3 Two-component spectrum model (TCM)

The two-component (soft+hard) model (TCM) for hadron spectra was first introduced in Ref. [5] for 200 GeV $p$-$p$ collisions. Given a $p$-$p$ spectrum TCM for unidentified-hadron spectra with soft and hard charge densities $\bar\rho_s$ and $\bar\rho_h$ [6], a TCM for PID hadrons can be generated by assuming that each hadron species $i$ comprises certain *fractions* of soft and hard TCM components denoted by $z_{si}$ and $z_{hi}$ (both $\leq 1$). The PID spectrum TCM can then be written as [8,9]

$$\bar\rho_{0i}(y_t, n_s) \approx d^2 n_{chi}/m_t dm_t dy_z \approx z_{si}(n_s)\bar\rho_s \hat{S}_{0i}(y_t) + z_{hi}(n_s)\bar\rho_h \hat{H}_{0i}(y_t), \tag{2}$$

where $\hat{S}_{0i}(y_t)$ and $\hat{H}_{0i}(y_t)$ are unit-normal model functions approximately independent of $n_{ch}$, total charge density $\bar\rho_0 \equiv n_{ch}/\Delta\eta = \bar\rho_s + \bar\rho_h$, and $n_s = \Delta\eta\bar\rho_s$ serves as an event-class index.

Figure 2 illustrates definition and physical interpretation of TCM model functions. The first panel shows $p_t$ spectra from 200 GeV $p$-$p$ collisions in relation to fixed soft-component model $\hat{S}_0(y_t)$ (curves) that describes the data asymptotic limit for $n_{ch} \to 0$ [5]. Data hard components are complementary to $\hat{S}_0(y_t)$ and are modeled by $\hat{H}_0(y_t)$. TCM hard components are *quantitatively predicted* by convoluting $p$-$p$ jet fragmentation functions (second panel) with a minimum-bias jet spectrum appropriate to the $p$-$p$ collision energy (third panel) [7]. Predictions (curves) compared to data hard components (points) are shown in the fourth panel.

Figure 3 shows PID TCM spectra (curves) compared to data spectra (the same points appearing in Fig. 1). Derivation of the PID spectrum TCM for 5 TeV $p$-Pb collisions is described in Refs. [8–10]. All data are described within their published *statistical* uncertainties. It is important to note that the TCM does not result from fits to individual spectra.

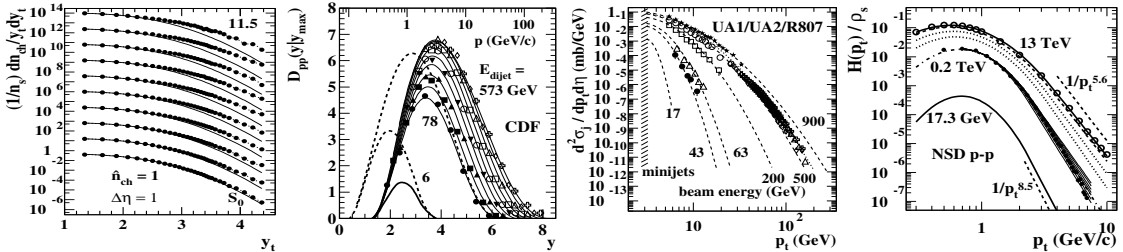

Figure 2: First: $y_t$ spectra for ten $n_{ch}$ classes of 200 GeV $p$-$p$ collisions. Second: Fragmentation functions for a range of jet energies from 2 TeV $p$-$\bar{p}$ collisions. Third: Jet energy spectra for several $p$-$p$ collision energies. Fourth: Predicted (curves) and measured (points) hard components for several $p$-$p$ collision energies.

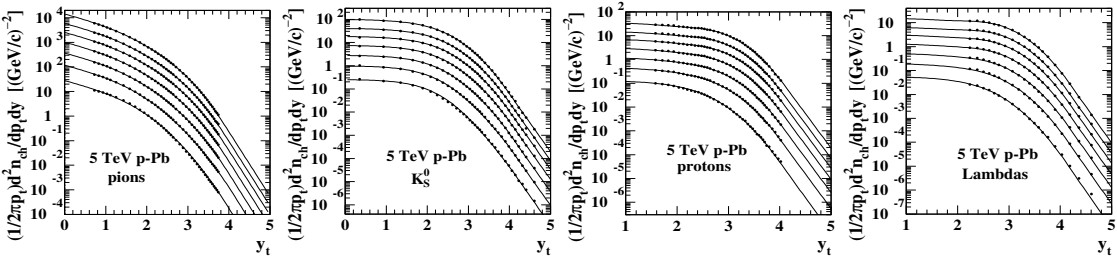

Figure 3: PID spectrum data from Fig. 1 (points) compared to a TCM description (curves) [9, 10]. The TCM is not fitted to individual spectra.

## 4 Spectrum shape evolution

Whereas the TCM provides absolute *predictions* for hadron yields as well as spectrum shapes, the BW model is not expected to provide such absolute predictions: "...the normalization of the spectrum...we will always adjust for a best fit to the data, because we are *only interested in the shape* of the spectra to reveal the dynamics of the collision zone at freeze-out [emphasis added]" [4]. "This [assumed collective hydrodynamic flow] results in a characteristic dependence of the [spectrum] shape which can be described with a common kinetic freeze-out temperature parameter $T_{kin}$ and a collective average expansion velocity $\langle \beta_t \rangle$ [citing Ref. [4]]" [3]. Is a BW model shape physically interpretable? Given that limitation one may elect to invoke model comparisons based on neutral shape measures so as to provide unbiased comparisons. "Neutral measure" here means a statistical measure motivated by effective statistical analysis via standard practice rather than by a sought-after result.

One possibility is logarithmic derivatives to determine *local spectrum curvature*. A logarithmic derivative is $(1/f)df/dx = d\ln f/dx$. The *second* derivative of the logarithm of spectrum $\bar{\rho}_0(y_t)$, $-d^2\ln[\bar{\rho}_0(y_t)]/dy_t^2$, approximates local curvature of the spectrum plotted in a semilog format. Since Fig. 3 and Refs. [9, 10] establish that the TCM is *statistically equivalent* to PID spectrum data, elements of the TCM can be used to explain curvature trends. For the TCM hard component, curvature is approximately a fixed value $1/\sigma_{y_t}^2$ (in terms of the $\hat{H}_0(y_t)$ Gaussian width) near its mode but falls to zero for the power-law tail at higher $y_t$. For the TCM soft component, curvature is approximated by $\propto \cosh(y_t)$ at lower $y_t$ since spectra (except for pions) are well-approximated there by a Boltzmann exponential on $m_t = m_i \cosh(y_t)$ [2].

Figure 4 (first) shows local curvatures vs $y_t$ for seven event classes of 5 TeV $p$-Pb collisions. As noted, curvature goes as $\propto \cosh(y_t)$ at lower $y_t$, rises toward a saturation value (hatched band) near the hard-component mode and then falls to zero for the power-law tail. Detailed centrality dependence corresponds to varying relative amplitudes of hard and soft components.

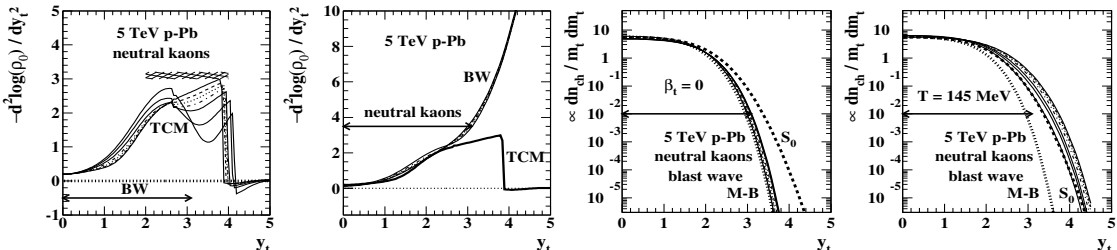

Figure 4: First: Local curvature trends for neutral kaons and for seven event classes of 5 TeV $p$-Pb collisions represented by the TCM. Second: Local curvatures for BW model fits to data demonstrating large deviations. Third: BW model curves corresponding to fixed zero radial flow $\langle \beta_t \rangle = 0$ that closely approximate a Boltzmann distribution (bold dotted curve). Fourth: BW model curves corresponding to fixed temperature $T = 145$ MeV compared to the soft component $\hat{S}_0(y_t)$ of the $p$-Pb TCM (bold dashed).

Figure 4 (second) shows the same procedure applied to BW model functions (curves of several line styles) as in Fig. 1. Included for reference is the TCM curve at left corresponding to the most-central event class (bold solid). There is a dramatic difference between TCM (statistically equivalent to data as noted) and BW model above $y_t = 2.7$ ($p_t \approx 1$ GeV/c). Ironically, the BW model has no sensitivity to detailed variation of *data* spectrum shapes.

Figure 4 (third) shows BW model functions corresponding to fit parameters from Ref. [3] except $\langle \beta_t \rangle \to 0$ (zero radial flow) in which case the curves approximate a Boltzmann exponential (bold dotted) which is consistent with the basic assumptions of Ref. [4]. The fourth panel shows BW model functions corresponding to fit parameters from Ref. [3] except temperature $T_{kin}$ is held fixed at 145 MeV. Comparison of third and fourth panels relative to fixed model $\hat{S}_0(y_t)$ demonstrates the main effect of nonzero $\langle \beta_t \rangle$ within the BW model. Note that the most-peripheral event class in the fourth panel (the lowest solid curve), with $\langle \beta_t \rangle \approx 0.25$, is consistent with TCM soft-component model $\hat{S}_0(y_t)$ (bold dashed) that represents the $n_{ch} \to 0$ limiting case (zero particle density). What mechanism generates flows at zero particle density?

## 5 Spectrum model fit quality

A standard measure of data-model fit quality is the Z-score (for an $i^{th}$ observation) [11]

$$Z_i = \frac{O_i - M_i}{\sigma_i}, \tag{3}$$

where $O_i$ is an observation, $M_i$ is a model prediction and $\sigma_i$ is the uncertainty for the observation. For an acceptable model one expects $Z_i$ to have an r.m.s. value near 1.

Figure 5 shows BW model Z-scores for pions, neutral kaons, protons and Lambdas, where uncertainties $\sigma_i$ are published *statistical* errors from Ref. [3]. Since $\chi^2 = \sum_i Z_i^2$ these results imply $\chi^2/\text{ndf} \sim O(25-100)$. Even within imposed fit intervals (arrows) *chosen to favor the model* the Z-scores are sufficiently large as to falsify the BW model as applied to these data.

Figure 6 shows TCM Z-scores for the same data with the same uncertainties. The TCM is applied to all available data over full $y_t$ acceptances (no restricted fit intervals) and *is not fitted to individual spectra*. Z-scores are consistent with an acceptable model: random $O(1)$ fluctuations with the exception of narrow excursions (for *charged* pions and protons) that *are* statistically significant but may be local data anomalies associated with $dE/dx$ PID [9].

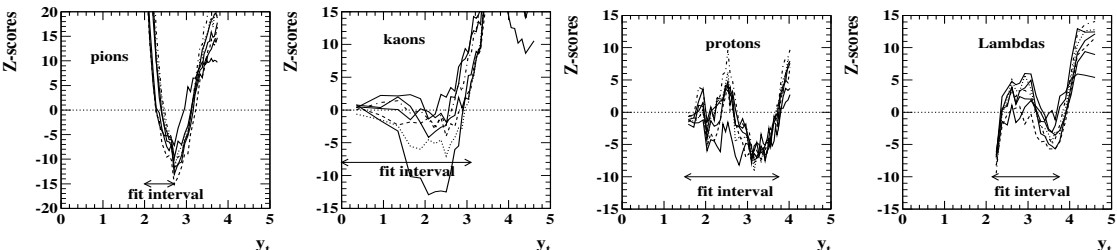

Figure 5: Z-scores for BW model fits to pion, neutral kaon, proton and Lambda spectra as in Fig. 1. Statistical uncertainties are used. Arrows indicate fit intervals.

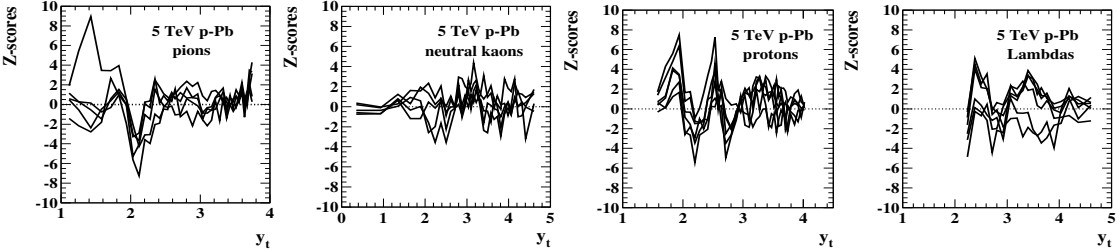

Figure 6: Z-scores for TCM applied to four hadron species. The TCM is not fitted to individual spectra. Note that kaon data are described accurately in $p_t \in [0,7]$ GeV/c.

## 6   What is an elementary collision?

The high-energy heavy ion program initially assumed that QGP formation *might* occur in more-central A-A collisions, but confirmation of that achievement required control (null) experiments in the form of *p-p*, *p*-A and *d*-A (small systems) data, i.e. hadron production from *elementary collisions* (e.g. "cold nuclear matter") where QCD is nominally well understood. Data manifestations of QGP formation in A-A should contrast dramatically (?) with data trends from elementary collisions. However, arguments based on certain correlation features (ridges) and *evolution* of hydrodynamic theory to achieve "good agreement with the data" assert that nominally "elementary" (small system) collisions actually support hydrodynamic evolution manifested by "flow-like features," [1] which begs the question: what is an elementary collision?

Figure 7 (first) shows a $\sqrt{s_{NN}} \approx 19$ GeV S-S pion spectrum (dots), the data that motivated Ref. [4], compared to a Boltzmann exponential (dash-dotted). Within the BW model context *any* deviations from a Boltzmann reference curve in the lab frame must indicate a boosted particle source: radial flow. Also plotted are data (open circles) from 17 GeV *p-p* collisions and TCM soft-component model $\hat{S}_0(m_t)$ (dashed, $T = 145$ MeV) appropriate for 19 GeV [7]. The second panel shows an $m_t$ spectrum from 91 GeV $e^+$-$e^-$ collisions with $q$-$\bar{q}$ dijet final state [12] compared to the same $\hat{S}_0(m_t)$. Does that mean there is radial flow in $e^+$-$e^-$ collisions?

Figure 7 (third) replots the same $e^+$-$e^-$ $m_t$ spectrum vs pion $y_t$ with $\hat{S}_0(m_t)$ now based on $T = 90$ MeV. $e^+$-$e^-$ data are described within uncertainties. The fourth panel presents an $e^+$-$e^-$ dijet longitudinal momentum spectrum as a density on $y_z$ (pion mass assumed). The point of this comparison is that hadrons, even from the most elementary $e^+$-$e^-$ collisions, follow a Boltzmann distribution *with power-law tail* $\hat{S}_0(m_t)$ on transverse momentum, a basic characteristic of high-energy collisions and parton or nucleon *fragmentation*. Deviations from a Boltzmann exponential cannot be used to claim emission from a flowing particle source.

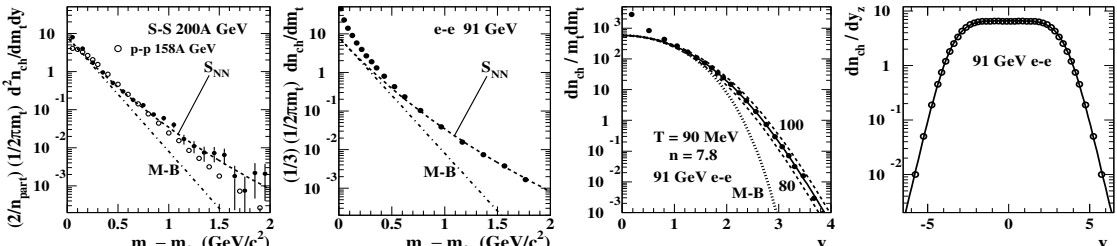

Figure 7: First: TCM soft-component $\hat{S}_0(m_t)$ compared to 19 GeV S-S data (solid points) and Boltzmann exponential (dash-dotted). Second: $m_t$ spectrum from 91 GeV $e^+$-$e^-$ collisions (points) [12] compared to $\hat{S}_0(m_t)$ with $T = 145$ MeV. Third: The same $e^+$-$e^-$ spectrum but with $T = 90$ MeV for $\hat{S}_0(m_t)$. Fourth: $e^+$-$e^-$ longitudinal momentum spectrum on rapidity with pion mass assumed [12].

## 7 Conclusion

Model-independent shape measures and Z-scores (based on statistical uncertainties) falsify the BW spectrum model from Ref. [4] as applied to 5 TeV $p$-Pb PID spectra from Ref. [3]. Monolithic BW model parameters, conventionally associated with radial flow, mimic TCM nonjet and jet contributions over limited $p_t$ intervals. The soft components of $p$-Pb $p_t$ spectra are consistent with $e^+$-$e^-$ dijet $p_t$ spectra. The data hard components are predicted by measured jet properties. It is certainly true of high-energy nuclear collisions that almost all high-$p_t$ hadrons are jet fragments. But it is equally true that almost all jet fragments are *low-$p_t$* hadrons. Spectrum models with a single component (monolithic), and especially with no jet description, cannot successfully model spectrum data, especially the strong low-$p_t$ jet fragment contribution with peak near 1 GeV/c. BW model fits thus cannot provide evidence for flows, "collectivity" or QGP in small collision systems. The parameter values may have no physical significance.

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
