# Peer review of "How the Blast-Wave Model Describes PID Hadron Spectra from 5 TeV p-Pb Collisions"

_SciPost Physics Proceedings, doi:SciPost Phys. Proc. 15, 025 (2024)_

## Round 1 · Referee Report · Anonymous (Referee 1) · 2023-1-18

Strengths

A coherent mini-summary of the authors's work in this area, making the case for a general audience and presenting a compelling comparison in which the TMD model appears to fit data much better than the BW explanations.

Weaknesses

Figures are crucial to the argument but too small to properly read. Make the figures significantly larger (the excess page count will not be a problem, as that primarily related to volume of text).

Report

Thank you for the short contribution. Coherent, readable, mostly accessible, and making a compelling case. Some of the points in Sec 6 seem perhaps antagonistically made toward BW models, but the arguments beneath are well motivated: perhaps calm down this section and the final conclusion line. It is a solid representation of the conference talk.

Requested changes

  1. The figure plots are very small and hard to read. It would be better to make them twice the linear size, laid out in 2 x 2 grids rather than four abreast. This is fine within the spirit of the page limit.

  2. Several places: convoluted -> convolved

  3. Top of p3: pt spectra -> yt spectra

  4. Sec 4: what are "neutral shape measures"?

  5. Sec 4: the definition and relevance of curvature, and why it (usually identified with a second derivative) can be accessed by a logarithmic derivative is not obvious: please clarify.

  6. Sec 4 end: this important most peripheral event class isn't obviously identifiable in the figure. As this seems a pertinent point, can it be more unambiguously identified?

  • validity: good
  • significance: good
  • originality: good
  • clarity: high
  • formatting: excellent
  • grammar: excellent

Author:  Thomas Trainor  on 2023-01-18  [id 3248]

(in reply to Report 1 on 2023-01-18)

  1. The figure plots are very small and hard to read. It would be better to make them twice the linear size, laid out in 2 x 2 grids rather than four abreast. This is fine within the spirit of the page limit.

Those figures all appear in cited Ref. [2] and other cited references in somewhat larger sizes and with more complete descriptions. This format is one I've used for conference proceedings for many years. See my previous ISMD proceedings articles.

Note that a single panel of a pair in Phys Rev double-column format is 1-1/4 inches wide. A single panel in this article is 1-1/16 inches wide. A pair of panels here is 5/6 the pair width in the PR format.

  1. Several places: convoluted -> convolved

For the text editor into which I am typing "convolved" is not recognized. "Convoluted" is suggested instead. In fact "convoluted" is the correct mathematical term. Convolution is a mathematical operation that maps two (or more, see the QCD factorization theorem) functions to a third via a convolution integral.

  1. Top of p3: pt spectra -> yt spectra

As noted in the text, what is plotted is "densities on pt vs pion transverse rapidity yt...." Also see the y-axis labels. Thus, what you see are pt spectra. The data y-axis values are numerically exactly the pt spectrum data published by ALICE. I introduced yt for spectrum plots more than fifteen years ago (see Ref. [5]) as a valuable improvement to spectrum visualization and have used it in some fifty publications since. It is notably hermetically shunned by a community that really doesn't want to know what its data are telling it.

  1. Sec 4: what are "neutral shape measures"?

The text says "...neutral shape measures so as to provide unbiased comparisons." There is a lamentable tendency in the field to apply some data measure (e.g. v2) or model (e.g. BW) associated with a desired outcome (i.e. non-neutral) and bend or break the rules of statistics to declare victory -- confirmation bias. The BW model is a notorious example -- see the first panel of Fig. 1 for an egregious example. The pion data loudly tell me that the BW model has no relevance to them.

  1. Sec 4: the definition and relevance of curvature, and why it (usually identified with a second derivative) can be accessed by a logarithmic derivative is not obvious: please clarify.

A logarithmic derivative has the form (1/f)df/dx = dlnf/dx. The expression in the text is the second derivative of lnf which gives the local curvature of lnf (e.g. spectrum data in a semilog plot) as an approximation (it is missing the first derivative in the proper curvature definition which is OK near an extremum). The main point is that the same "neutral shape measure" -- an approximation to local curvature -- applied to TCM (statistically equivalent to data as noted, see Fig. 6) and BW gives dramatically different results: i.e. one way of demonstrating that BW is falsified by data.

Action: Some text to clarify should be inserted.

  1. Sec 4 end: this important most peripheral event class isn't obviously identifiable in the figure. As this seems a pertinent point, can it be more unambiguously identified?

It is basic to the BW model that boost or beta-t increases with centrality, pushing the model out to higher pt. Thus, the most peripheral BW instance is the lowest curve in the fourth panel which coincides with the bold dashed TCM soft component S0 that describes data in the limit nch goesto 0 -- zero particle density -- as noted in the text. Also, compare third and fourth panels with zero and nonzero beta-t respectively. Those panels appear as Fig. 12 in Ref. [2] where more details are provided. That is why the BW model applied to low nch p-p data always returns beta-t approx 0.25 which is nonsense. Then see "The parameter values may have no physical significance."

Action: Equivalent text (without the pejorative) should be added to clarify.

You note: "Some of the points in Sec 6 seem perhaps antagonistically made toward BW models, but the arguments beneath are well motivated: perhaps calm down this section and the final conclusion line. It is a solid representation of the conference talk."

I have just reread Sec. 6 and find no errors of fact. It reads as I intended. The imposition of BW models on small collision systems and interpretation of any forthcoming results as indicating flows demonstrably constitutes pseudoscientific gibberish. That is my opinion as an independent scientist with good math skills based on several decades of study in this field. My treatment in the text is actually understated compared to what is warranted.

You note: "...the TMD model appears to fit data much better than the BW explanations."

To clarify, the two-component model or TCM is not fitted to individual data. It is predictive and applies broadly to multiple collision systems with a few smooth parameter variations related to fundamental QCD processes. I suspect you meant to say "to describe data much better" which is true.

Action: In the caption of Fig. 2 the word "CHECK" was left in by mistake and should be removed.

Thanks for the thoughtful review.

---

## Editorial Decision

published